# Relationship Between Systemic Inflammatory Response Exponents, Levels of ADAM10, ADAM17 Proteins and Selected Clinical Parameters in Patients with Colorectal Cancer: Original Research Study

**DOI:** 10.3390/ijms26031104

**Published:** 2025-01-27

**Authors:** Magdalena Sikora-Skrabaka, Katarzyna Weronika Walkiewicz, Dariusz Waniczek, Joanna Katarzyna Strzelczyk, Ewa Nowakowska-Zajdel

**Affiliations:** 1Department of Nutrition Related Prevention, Department of Metabolic Diseases Prevention, Faculty of Public Health in Bytom, Medical University of Silesia in Katowice, 41-902 Bytom, Poland; magda_sikora19@wp.pl; 2Department of Clinical Oncology, No. 4 Provincial Specialist Hospital, 41-902 Bytom, Poland; 3Department of Internal Diseases Propaedeutics and Emergency Medicine, Faculty of Public Health in Bytom, Medical University of Silesia, 41-902 Bytom, Poland; katarzyna.walkiewicz@sum.edu.pl; 4Department of Oncological Surgery, Faculty of Medical Sciences in Zabrze, Medical University of Silesia, 40-515 Katowice, Poland; dwaniczek@sum.edu.pl; 5Department of Medical and Molecular Biology, Faculty of Medical Sciences in Zabrze, Medical University of Silesia in Katowice, 41-808 Zabrze, Poland; jstrzelczyk@sum.edu.pl

**Keywords:** colorectal cancer (CRC), ADAM10, ADAM17, proteins, SIR, NLR, PLR, inflammation, diabetes mellitus type 2 (DMT2), cardiovascular diseases (CVD)

## Abstract

Chronic inflammation is a confirmed risk factor for colorectal cancer (CRC). Indicators of systemic inflammatory response (SIR), such as neutrophil-to-lymphocyte ratio (NLR) or platelet-to-lymphocyte ratio (PLR), are easily accessible indicators of the generalized inflammatory response. At the molecular level, inflammation-related carcinogenesis involves proteins from the adamalysin family: ADAM10 and ADAM17. The aim of the study was to assess NLR and PLR and their relationship with selected clinical parameters in CRC patients, as well as the correlation between ADAM10 and ADAM17 in tumor tissue and matched surgical margins with NLR and PLR values. Tumor tissue material matched surgical margins, and blood was collected from 66 patients who underwent surgery because of CRC. The concentrations of ADAM10 and ADAM17 in the collected material were tested using the enzyme-linked immunosorbent assay (ELISA) method. SIR parameters (NLR, PLR) were also determined. The results were statistically analyzed and compared with selected clinical parameters. Results: The study showed that PLR was lower in patients with comorbid cardiovascular diseases (CVD). In patients who underwent preoperative treatment, both the NLR and PLR values were higher than in patients who underwent primary surgery. There was also a negative correlation between ADAM17 concentrations in the surgical margin and PLR values. In conclusion, the presence of additional diseases such as CVD or diabetes mellitus type 2 (DMT2) or the use of preoperative treatment should be taken into account when assessing SIR parameters in CRC patients. Moreover, no clear correlations have been found between ADAM10 and ADAM17 and SIR parameters.

## 1. Introduction

Chronic inflammation constitutes a well-established risk factor for the development of cancer, including colorectal cancer (CRC). This observation is confirmed by the fact that patients suffering from inflammatory bowel diseases, such as Crohn’s disease or ulcerative colitis, are at increased risk of developing CRC [1]. The relationship between the systemic inflammation and the local immune response is also known and has a documented basis in the pathogenesis not only of CRC but also of many other cancer diseases [2]. In recent years, the relationship between inflammation and cancer has attracted the attention of researchers. Indicators of systemic inflammatory response (SIR), such as neutrophil-to-lymphocyte ratio (NLR) or platelet-to-lymphocyte ratio (PLR), are related to tumor size and prognosis in CRC [3], but also in other malignant tumors, such as lung cancer [4], breast cancer [5] or esophageal cancer [6]. One study found that high PLR and high NLR are independent factors of poor prognosis in terms of overall survival (OS) and disease free survival (DFS) in patients with CRC [7]. The relationship between SIR rates and the response of CRC patients to treatment has been investigated. Low baseline NLR levels and early NLR reduction were significantly associated with better prognosis in metastatic CRC (mCRC) patients treated with immunotherapy [8]. According to the most recent observations, the risk of future cancer can be determined based on SIR indicators. Based on the British Biobank cohort, a positive correlation of NLR and PLR with the risk of seven cancers (including CRC) was observed [9]. All recent meta-analyses agree on the important role of SIRs as both predictive and prognostic factors in CRC [10,11].

At the molecular level, proteins from the ADAM and TNF families are involved in the inflammatory mechanisms of CRC carcinogenesis. ADAM10 and ADAM17 are proteins that have been proven to be related to inflammation. ADAM17 has a documented role in the pathogenesis of CRC by influencing angiogenesis, secretion of proinflammatory cytokines (IL6, IL10) [12], and activation of growth factors from the EGF family [13]. Moreover, ADAM17 stimulates the development of inflammation by influencing TNF alpha [14,15]. On the other hand, ADAM10 modulates inflammatory response pathways through Notch-related signaling [16]. Many studies confirm the involvement of ADAM10 and ADAM17 in the pathogenesis of CRC, as well as their relationship with the stage of advancement, metastasis and response to treatment [13,16,17,18].

In a previous study, we analyzed the concentration of ADAM10 and ADAM17 in the tumor tissue and surgical margin in patients with CRC and the role of these proteins in other diseases dependent on the inflammatory process, such as DMT2 or CVD [19]. In order to proceed with our research, in this manuscript we focused on the assessment of SIR parameters such as NLR, PLR and their relationship with selected clinical parameters, including comorbidities in patients with CRC, as well as examining the correlation between ADAM10 and ADAM17 concentrations in tumor tissue and matched surgical margins with NLR and PLR values. Additionally, the correlation between the concentration of ADAM10 and ADAM17 in blood serum and tissue material collected from the tumor and the matched surgical margin in patients operated on for CRC was investigated.

## 2. Results

### 2.1. Characteristics of the Study Groups

#### 2.1.1. The Main Study Group

The main study group consisted of 66 patients who underwent surgery for CRC. The analysis included ADAM10 and ADAM17 concentrations in the tumor tissue and matched surgical margin, as well as selected clinical and laboratory parameters, such as TNM stage according to the 8th edition [20], the presence of selected comorbidities and SIR parameters: NLR and PLR. The characteristics of the study group are presented in Table 1. The analysis of ADAM10 and ADAM17 concentrations in the tumor tissue and surgical margin in CRC patients and their analysis, depending on selected parameters, were presented in our previous study [19]. Regardless of the above, in Table 2, we present the concentrations of ADAM10 and ADAM17 in selected groups of patients.

#### 2.1.2. The Additional Study Group

An additional study group included selected patients from whom both tissue material and a blood serum sample were obtained for testing (n = 24). In this group, the concentrations of ADAM10 and ADAM17 were determined in both tissue material and blood serum. Due to the small size of this group, the presence of occurring diseases or the location of the primary tumor was not taken into account, and the clinicopathological stage was divided into advanced disease (stage IV) and earlier stages (stage I-III). The characteristics of the study group are presented in Table 3. Individual ADAM10 and ADAM17 concentration results are presented in Table 4. There was a lower serum concentration of ADAM10 in women than in men (67.9 (49.8–89.5) vs. 171.4 (93.25–227.55); *p* = 0.007).

### 2.2. SIR Exponents in Patients with CRC and Their Correlation with ADAM10 and ADAM17 and Selected Clinical Parameters

The results of the SIR parameters NLR and PLR for the main study group are presented in Table 5.

In patients with concomitant CVD, PLR values were significantly lower than in patients without CVD (136.72 ± 67.67 vs. 228.77 ± 127.75; *p* = 0.007), while for NLR no statistical significance was demonstrated. In the group of patients with DMT2, both NLR (2.7 (2.13–4.61) vs. 3.43 (2.34–4.85); *p* = 0.34) and PLR (157.14 (129.23–219.88) vs. 217.2 (144.33–268.06); *p* = 0.07) had lower values than in patients without DMT2, and no significant difference was found. Moreover, in patients who underwent preoperative treatment—radiotherapy (RT) or radiochemotherapy (RCT) before surgery, both NLR (4.0 (3.7–4.71) vs. 2.78 (2.14–4.68); *p* = 0.041) and PLR (292.48 ± 195.5 vs. 198.8 ± 105.15; *p* = 0.05) were higher than in patients who initially underwent surgery. No statistically significant differences were found for the NLR and PLR values depending on the location of the colorectal tumor (right or left side), as well as depending on the clinicopathological stage, tumor size (T feature) or the presence of metastases in the lymph nodes (N feature).

Correlations between ADAM10 and ADAM17 concentrations in the tumor tissue and surgical margin, as well as NLR and PLR values, were also examined. There was a weak negative correlation between ADAM17 concentrations in the surgical margin and PLR values (R = −0.27; *p* = 0.032)—Figure 1. For the rest, no statistically significant correlations were found. Figures showing the remaining correlations are included in the Appendix A.

### 2.3. ADAM10 and ADAM17 Concentration in Tissue and Blood Serum of Patients with Colorectal Cancer

The focus was on the analysis of correlations between the concentrations of selected adamalysins in tissue material and patients’ blood serum. A moderate positive correlation was found between the ADAM10 concentration in the tumor tissue and the ADAM10 concentration in the surgical margin (r = 0.73; *p*< 0.001)—Figure 2. Similarly, a small positive correlation was found between ADAM17 concentration in tumor tissue and ADAM17 concentration in margin tissue (R = 0.4; *p* = 0.04)—Figure 3.

There was no correlation between the ADAM10 concentration in patients’ serum and the ADAM10 concentration in the tumor sample (R = 0.1; *p* = 0.64) and in the margin sample (R = 0.13; *p* = 0.58). Similarly, no correlation was found between the ADAM17 concentration in patients’ serum and the ADAM17 concentration in the tumor sample (R = 0.02; *p* = 0.92) and in the margin sample (R = 0.05; *p* = 0.82). Figures showing the remaining correlations are included in the Appendix A.

Additionally, multiple regression analysis was performed for ADAM 10 and ADAM17 concentrations in tumor tissue, margins and blood serum, assessing the relationship for individual stages of CRC advancement, and no relationships were found.

## 3. Discussion

In the presented study, in accordance with the assumed goal, it was possible to demonstrate the existence of a relationship between SIR parameters and selected clinical features. However, when examining the correlations between SIR parameters and the concentrations of ADAM10 and ADAM17, only a relationship was found between the concentration of ADAM17 in the surgical margin and the PLR value in the form of a weak negative correlation (Figure 1).

The main aim of the study was the analysis of SIR parameters (NLR and PLR) and their correlation with selected clinical and pathomorphological parameters. The following factors were taken into account: the clinical and pathological stage of advancement, the location of the primary tumor, preoperative treatment or lack thereof in patients with rectal cancer. As for comorbidities, the analysis included the history of DMT2 and CVD due to the fact that these diseases most often develop on the basis of inflammatory lesions [21,22].

The most interesting observation of the study is the fact that the tested SIR rates (NLR, PLR) were lower in CRC patients with CVD and DMT2, but only PLR in CRC patients with comorbid DMT2 achieved statistical significance. Chronic inflammation is a well-known factor in the pathogenesis of DMT2 and CVD [21,22]. DMT2 and insulin resistance are associated with the overexpression of many inflammatory cytokines, such as tumor necrosis factor alpha (TNF-alpha), IL1 or IL6 [23], which may affect the blood vessels and lead to CVD. Researchers also draw attention to the link between obesity (and the associated inflammation) and metabolic diseases, including CVD and DMT2 [24]. Proteins from the adamalysin family (ADAM10 and ADAM17) also have a documented role in these diseases due to their involvement in inflammation [25,26]. We did not find any studies in which the presence of such comorbidities and direct assessment of how such diseases could affect the value of SIR parameters were considered when assessing SIR parameters in CRC patients. It can be assumed that in patients who previously had such diseases associated with the activation of the systemic inflammatory response when an additional cancer disease (e.g., CRC) develops, the inflammatory response is not as pronounced as in patients without previous chronic activation of the inflammatory system. Mertoglu and Gunay, in their study, showed that “NLR increases significantly in patients with prediabetes and diabetes, while PLR decreases in prediabetes and early stages of diabetes, but increases in later stages.” [27]. We also know from other studies that high NLR and PLR values correlate with diabetes complications, such as diabetic foot [28], diabetic retinopathy [29] or other diabetic microvascular complications [30]. The role of SIR indicators in predicting the occurrence of cardiovascular and metabolic diseases [31] and their prognosis is also confirmed [32].

Furthermore, the study showed that in patients after neoadjuvant treatment both NLR and PLR values were higher than in patients who underwent surgery without neoadjuvant treatment. Researchers assume that as a result of systemic treatment (RT or RCT), the body’s inflammatory response is more activated. However, this observation requires further research on larger groups of patients. No studies were found assessing the difference in SIR parameter values in patients treated preoperatively compared to patients undergoing primary surgery. However, there are reports suggesting that a high PLR in patients after preoperative chemotherapy correlates with a reduced objective response rate (ORR) [33]. In recent years, researchers have been paying a lot of attention to examining the impact of both the pretreatment SIR values themselves [11,34,35] and the difference (delta) in the values of these indicators in assessing the prognosis of CRC patients [36], as well as factors of response to treatment [37]. In the current study, the authors focused only on simple correlation analyses, but a longer observation period and more in-depth analyses will also allow us to examine the assessment of SIR parameters in our study group on the prognosis of patients.

NLR values for healthy adults have been estimated to be between 0.78 and 3.53 (mean NLR value 1.65 ± 1.96) [38], while the most frequently reported cut-off point for NLR for CRC patients is 2–5 [39]. In our study group of CRC patients, the NLR value was 3.1 (2.18–4.69). Available studies assessing the correlation of NLR with the stage or size of the tumor showed that the NLR value is higher in stages III/IV compared to stage I/II, and that the NLR value increases with the increase in clinicopathological advancement [40,41]. However, some of them did not show any significant correlation in this area [42]. The literature has also shown an association with NLR and lymph node metastases [43].

In the current study, NLR values are the lowest for patients with stage I and the highest for patients with stage IV; however, no statistically significant differences were found between individual stages of advancement. The highest NLR values for patients with the most advanced disease can be explained by the fact that as the disease progresses, the number of neutrophils increases, as an indicator of the inflammatory response to tumor cells, and at the same time, the cellular response may decrease, as indicated by a decrease in the number of lymphocytes. The lack of a statistically significant difference in the study may be caused by too small patient subgroups.

While NLR has documented significance in CRC patients, the value of PLR is still unclear. Summarizing the results of recent studies, the average cut-off value for PLR in CRC patients was 146.98 and depended on the tumor size and stage of cancer [36]. For our study group, this value was slightly higher and amounted to 182.89 (137–250). Some studies show that the PLR value is related to the size and degree of tumor invasion [44] or a higher PLR value was demonstrated for stage IV compared to stages II-III [45]. On the other hand, there are reports according to which PLR is not related to the advancement of CRC [46].

The results of our analysis show that the PLR value is higher in CRC stages III and IV than in earlier stages, but the differences are not statistically significant. The increase in PLR with increasing stage is related, firstly, to the increased production of thrombopoietin (TPO) and IL-6 by tumor cells and, secondly, to the fact that tumor cells directly stimulate platelets.

ADAM10 and ADAM17 are proteins involved in carcinogenesis by activating pathways related to the inflammatory process, including TNF-alpha, insulin-like growth factors (IGFs), vascular endothelial growth factor (VEGF), fibroblast growth factor (FGF) and epidermal growth factor (EGF) [14,47]. Therefore, in the current study, we decided to evaluate whether there is a correlation between SIR indices and the concentration of ADAM10 and ADAM17 in tumor tissue and surgical margin. In our study, we presented a weak negative correlation between ADAM17 concentrations in the surgical margin and PLR values. Only one study was published in the literature, examining the association between ADAM17 and NLR. In animal models with SARS-CoV-2 infection, inhibition of ADAM17/MMP reduced NLR value [48]. However, no studies on the relationship between ADAM17 and the PLR value were found in the PubMed database.

In available literature, we did not find any other studies presenting the assessment of ADAM10 and ADAM17 concentrations in tissue collected from the tumor and surgical margin as well as in blood serum from colorectal cancer patients in the same patients. Our previous study showed that the serum concentration of ADAM10 in patients with CRC was higher than in patients without CRC [49]. Another study showed that the concentrations of ADAM10 and ADAM17 in the tumor tissue and surgical margin of CRC patients differ [19]. In the current study, a positive correlation was shown between both ADAM10 and ADAM17 concentrations in the tumor tissue and the surgical margin. This might mean that the concentration of ADAM10 and ADAM17 directly in the tumor tissue and the surrounding tissue of the surgical margin is more closely correlated than with the concentration of these proteins in the serum. Hypothetically, the concentration of ADAM 10 and ADAM17 in serum will change later than it will in the tumor or surgical margin tissue. Further research on this topic in a larger group of patients may bring interesting results. However, these results may be distorted due to the too small study population. No correlation was found between the concentration of ADAM10 and ADAM17 in blood serum, tumor tissue and surgical margin.

## 4. Materials and Methods

### 4.1. Study Population

The study population consisted of 66 patients (32 men, 34 women) with CRC, who underwent elective colorectal surgery in the Department of General and Gastroenterological Surgery with the Oncological Surgery Branch, Bytom, Poland. In all cases, tumor samples and macroscopically unchanged surgical margins of approximately 1 cm^3^ were collected by the same colorectal surgeon. The surgical margin samples were verified histopathologically as free of cancer. The samples were transported to the laboratory on ice, and then the tissue pieces were washed with cold PBS buffer and weighed for the preparation of 10% tissue homogenates. In some patients, blood was also collected to test the concentration of ADAM10 and ADAM17 in serum. In addition, comorbidities, age and sex were analyzed. In patients with rectal cancer, preoperative treatment—RT or RCT—was taken into account. The results of selected laboratory tests were also collected, including peripheral blood count, and the NLR and PLR were assessed. All laboratory tests were performed before the surgery. Data were obtained from the patients’ medical records. Patients were divided into four main groups according to the TNM Classification of Malignant Tumours, 8th, based on clinical data, results of imaging studies and histopathological reports [20]. Patients with chronic inflammatory conditions (including inflammatory bowel diseases), previous cancer history, and those undergoing chronic immunosuppressive treatment were excluded from the study. The study was approved by the Ethics Committee of the Medical University of Silesia (14 July 2020)—number PCN/0022/KB1/42/VI/14/16/18/29/20. Written informed consent was obtained from all participants.

### 4.2. Total Protein Concentrations in Tissue Samples

The first stage was the preparation of 10% tissue homogenates. For this purpose, a tissue fragment (30–60 mg) was ground in a PRO 200 mechanical homogenizer (PRO Scientific Inc, Oxford, CT, USA) at a speed of 10,000 RPM (5 times for 1 min at 2-min intervals) in the presence of an appropriate volume of cooled PBS buffer (PBS without Ca and Mg; pH 7.4; BIOMED, Lublin, Poland) containing 0.5% Triton^®^ X-100 (Sigma-Aldrich, St. Louis, MO, USA). Then, the obtained homogenates were centrifuged at 4000 RPM for 15 min at +4 °C. The supernatants were divided into appropriate portions and frozen at −80 °C until further determinations were made. The AccuOrange ™ Protein Quantitation Kit (Biotium, Fremont, CA, USA) was used for total protein quantification. The determinations were made in tissue lysates according to the protocol attached to the kit, preparing 26-fold dilutions of samples for protein content measurement. The detection range was 0.1–15 µg/mL protein. Fluorescence was measured at an excitation wavelength of 480 nm and an emission wavelength of 598 nm (SYNERGY H1 microplate reader; BIOTEK, USA, using the GEN5 program). The determinations were made in duplicate. Determinations of total protein concentration were performed due to the need to express the concentration of ADAM10 and ADAM17 in units per µg of protein.

### 4.3. Determination of ADAM10 Concentration in Tissue Homogenates

The concentration of ADAM10 in tissue homogenates was determined using the ELISA method using the 96 tests for a disintegrin and metalloprotease 10 (ADAM10 assay kit, Cloud-Clone Corp., Katy, TX, USA) cat. no. SEA766Hu in accordance with the manufacturer’s instructions with a sensitivity of 28 pg/mL. Plates were read by Bio-Tek µQuant Universal Microplate Spectrophotometer (Bio-Tek, World Headquarters, Riverside, CA, USA), using 450 nm as the primary wavelength. Data Analysis Software KC Junior version 1.41.8 (Bio-Tek, Winooski, VT, USA) was used. All standards and samples were run in duplicates. The absorbance was transformed to concentration pg/mL.

### 4.4. Determination of ADAM17 Concentration in Tissue Homogenates

The concentration of ADAM17 in tissue homogenates was determined by the ELISA method using the Cloud-Clone Corp., (Katy, TX, USA) ELISA Kit ADAM17 (cat. no. SEB555Hu), 96 tests for a disintegrin and metalloprotease 17, in accordance with the instructions attached to the kit with a sensitivity of 0.059 ng/mL. The absorption was read at a wavelength of 450 nm using an ELISA µQuant reader (Bio-Tek, World Headquarters, Riverside, CA, USA), and the results were processed using the KCJunior Software version 1.41.8 (Bio-Tek, Winooski, VT, USA). The substance concentration value was read from the curve plotted for ADAM17 concentration standards expressed in ng/mL. The determinations were made in duplicates.

The concentrations of ADAM10 and ADAM17 in tissue material were determined in 30-fold dilution.

In the ADAM 10 assay kit, the microplate was precoated with an antibody specific to ADAM 10, for the ADAM 17 assay kit, with an antibody specific to ADAM 17. Samples and standards were added to microplate wells with a biotin-conjugated antibody specific to ADAM 10 or ADAM 17. Then, avidin conjugated to horseradish peroxidase (HRP) was added and incubated. Next, 3,3′,5,5′-Tetramethylbenzidine (TMB) substrate solution was applied. Avidin-HRP was used for detecting biotinylated antibodies.

### 4.5. Determination of ADAM10 and ADAM17 Concentrations in Blood Samples

During hospitalization, 3 cm^3^ blood samples were collected from patients, from which serum was obtained. Serum samples were frozen at −80 °C. The tests were performed using the ELISA KIT test (Cloud Clone Corporation) in accordance with the manufacturer’s recommendations. We used sets of well plates coated with antibodies specific for ADAM10 and ADAM17. We used biotin-avidin and horseradish peroxidase. Color changes of the tested solution were measured spectrophotometrically using light with a wavelength of 450 nm ± 10 nm. Adamalysin concentrations were determined by comparing the obtained results with those of the standard sample. To determine the concentrations of ADAM10 and ADAM17 in blood serum, the same kits were used for tissue material, but the determinations were made without diluting the samples.

All immunoassays were performed at the Department of Medical and Molecular Biology, Faculty of Medical Sciences in Zabrze, Medical University of Silesia in Katowice.

### 4.6. Statistical Analysis

Two subgroups were distinguished from the entire study group. The first group included all patients who had peripheral blood count results available (N = 66). The second subgroup consisted of patients who managed to collect both serum and tissue material from the tumor and matched surgical margins for the determination of ADAM10 and ADAM17 concentrations (N = 24). In all study patients, selected clinical, anthropometric and laboratory parameters were analyzed, such as the stage of cancer, tumor location and coexistence of diseases. Nominal and ordinal data were expressed as numbers (n), and interval data were expressed as mean ± standard deviation (SD) in the case of a normal distribution and as median (lower quartile–upper quartile) in the case of data with a skewed or non-normal distribution. The distribution of variables was assessed using the Shapiro–Wilk W test. To compare data between two groups, the Student’s *t*-test for independent groups was used in the case of normal distribution or the Mann–Whitney U test in other cases. ANOVA tests were used to compare multiple groups. The correlations were tested using Pearson or Spearman correlation coefficients, respectively. Statistical significance was set at *p* < 0.05. The statistical power was 0.8. Analysis was performed using Statistica version 13.3.

### 4.7. Research Hypothesis and the Aim of the Study

The concentration of ADAM10 and ADAM17 proteins in CRC patients depends on the activation of chronic inflammation (expressed by SIR indicators) and selected clinical parameters (comorbidities, sex, clinical and histopathological stage).
The main purpose of the study was to assess the SIR parameters (NLR, PLR) and their correlation with selected clinical parameters, as well as the correlation between ADAM10 and ADAM17 concentrations in tumor tissue and matched surgical margins with NLR and PLR values in patients with CRC.Additionally, we aimed to investigate correlations between the concentration of ADAM10 and ADAM17 in blood serum and tissue material collected from the tumor and matched surgical margin after elective surgery for CRC.

## 5. Conclusions

SIR parameters did not differ between CRC patients’ groups with different clinical features connected to cancer. On the other hand, lower PLR values were observed in CRC patients with a history of CVD. The impact of the history of DMT2 on the SIR parameters and the history of CVD on the NLR value requires confirmation because the differences did not reach statistical significance. Indisputably, NLR and PLR values were higher in CRC patients, who underwent neoadjuvant treatment before surgery. Moreover, a higher concentration of ADAM17 protein in the tissue from surgical margins was associated with lower PLR values. It is obvious that the correlations between selected SIR parameters, ADAMs and clinical features may also be influenced by factors other than those analyzed.

The analysis of the additional study group did not show any correlation between the concentrations of ADAM10 and ADAM17 in tissue material and blood plasma. However, it was shown that a higher concentration of ADAM10 and ADAM17 proteins in the tissue from the resection margin correlates with a higher concentration of ADAM10 and ADAM17 proteins in the tissue from the tumor.

## 6. Limitations

The authors are aware of the limitations of this study, such as the small sample size of the study group. We are also aware that the correlations between selected parameters may also be biased by factors other than those analyzed. Further research on the relationship between adamalysin concentrations in serum and direct tissue material is certainly recommended.

## Figures and Tables

**Figure 1 ijms-26-01104-f001:**
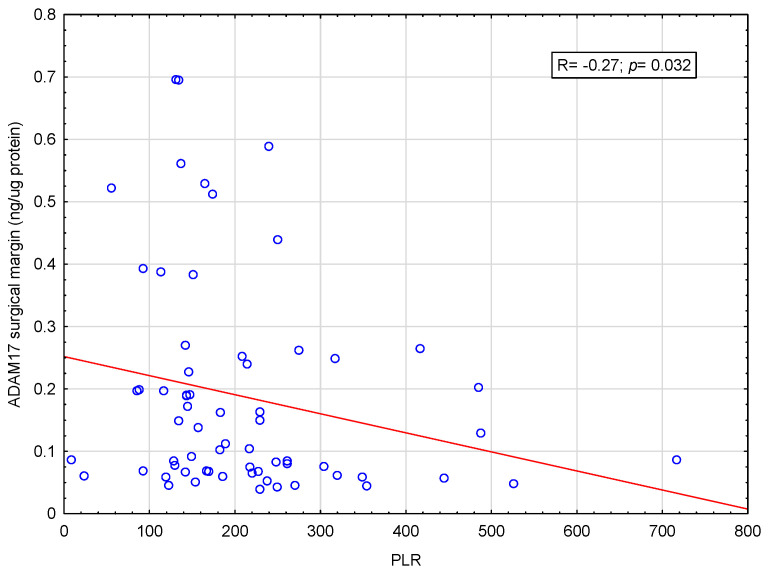
Correlation between PLR and ADAM17 concentration in surgical margin tissue. Demonstrated weak negative correlation between ADAM17 concentrations in the surgical margin and PLR values (R = −0.27; *p* = 0.032).

**Figure 2 ijms-26-01104-f002:**
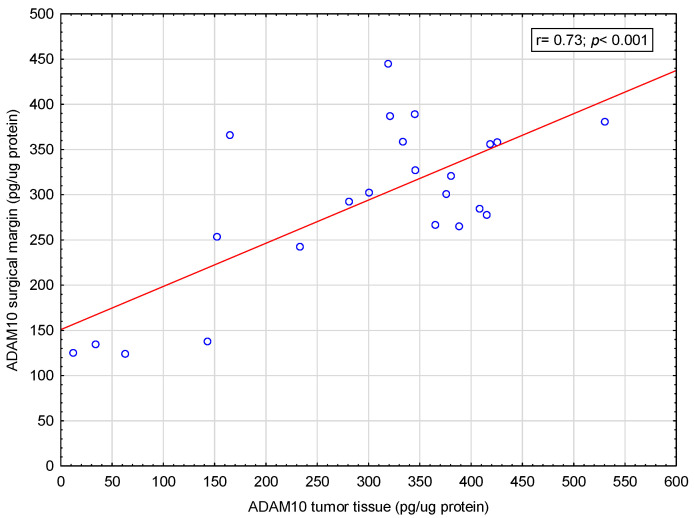
Correlation between ADAM10 concentration in tumor tissue and ADAM10 in the surgical margin. Demonstrated moderate positive correlation between ADAM10 concentration in tumor tissue and ADAM10 in the surgical margin (r = 0.73; *p* < 0.001).

**Figure 3 ijms-26-01104-f003:**
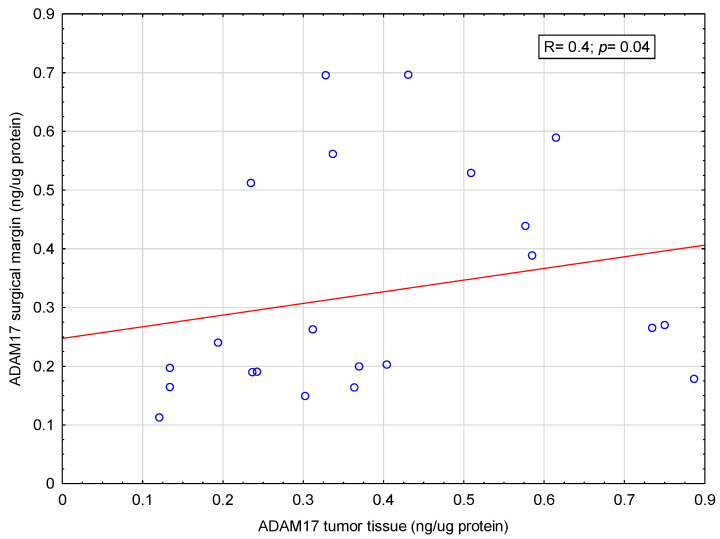
Correlation between ADAM17 concentration in tumor tissue and ADAM17 in the surgical margin. Demonstrated a slight positive correlation between ADAM17 concentration in tumor tissue and ADAM17 in the surgical margin (R = 0.4; *p* = 0.04).

**Table 1 ijms-26-01104-t001:** Characteristics of the main study group. Data are expressed as mean ± SD (standard deviation).

Characteristics	All Patients	Male	Female
n	66	32	34
Age (years)	67 ± 7	65 ± 10	68 ± 10
I stage (n)	12	5	7
II stage (n)	14	8	6
III stage (n)	20	7	13
IV stage (n)	20	12	8
DMT2 (n)	29	16	13
CVD (n)	43	23	20

n—number; DMT2—diabetes mellitus type 2; CVD—cardiovascular diseases.

**Table 2 ijms-26-01104-t002:** ADAM10 and ADAM17 concentrations according to selected clinical features in the main study group. Data are expressed as mean ± SD.

Characteristic	ADAM 10		ADAM17	
	Tumor (pg/μg protein)	Surgical margin (pg/μg protein)	Tumor (ng/μg Protein)	Surgical margin (ng/μg Protein)
All patients (n = 66)	232.11 ± 113.72	235.88 ± 81.24	0.23 ± 0.18	0.19 ± 0.17
Female (n = 34)	220.49 ± 108.44	243.7 ± 81.92	**0.22 ± 0.15 ***	**0.14 ± 0.13**
Male (n = 32)	244.1 ± 119.44	264.38 ± 80.48	0.24 ± 0.21	0.24 ± 0.19
I stage (n = 12)	193.73 ± 117.72	274.1 ± 64.73	0.17 ± 0.1	0.12 ± 0.05
II stage (n = 14)	206.64 ± 121.81	278.05 ± 76.61	0.24 ± 0.18	0.25 ± 0.21
III stage (n = 20)	256.3 ± 113.53	253.71 ± 103.71	0.25 ± 0.23	0.18 ± 0.14
IV stage (n = 20)	249.67 ± 103.57	226.23 ± 62.31	0.23 ± 0.17	0.19 ± 0.2
DMT2 (n = 29)	241.78 ± 124.54	250.12 ± 77.68	0.28 ± 0.23	0.22 ± 0.2
CVD (n = 43)	245.23 ± 116.58	258.75 ± 78.74	0.26 ± 0.21	0.21 ± 0.2

n—number; DMT2—diabetes mellitus type 2; CVD—cardiovascular diseases; Data with statistical significance are marked in bold. * *p* < 0.05: Higher ADAM17 concentration in tumor tissue than in surgical margin tissue in females (*p* = 0.03). Other results were not statistically significant (*p* > 0.05).

**Table 3 ijms-26-01104-t003:** Characteristics of the additional study group. Data are expressed as mean ± SD.

Characteristics	All Patients	Male	Female
n	24	12	12
Age (years)	71.92 ± 7.05	71.17 ± 6.3	72.67 ± 8
I stage (n)	4	2	2
II stage (n)	6	3	3
III stage (n)	8	3	5
IV stage (n)	6	4	2
DMT2	14	9	5
CVD	16	10	6

n—number; DMT2—diabetes mellitus type 2; CVD—cardiovascular diseases.

**Table 4 ijms-26-01104-t004:** ADAM10 and ADAM17 concentrations in serum, tumor tissue and surgical margin tissue in the additional group. Data are expressed as mean ± SD for parametric data or as median (lower quartile–upper quartile) for nonparametric data. Means were compared using the Student’s *t*-test for normally distributed samples and the Mann–Whitney U-test for non-normally distributed samples.

Characteristic	ADAM 10			ADAM17		
	Tumor (pg/μg protein)	Surgical margin (pg/μg protein)	Blood serum (pg/mL)	Tumor (ng/μg protein)	Surgical margin (ng/μg protein)	Blood serum (ng/mL)
All patients (n = 24)	285.17 ± 140.87	291.1 ± 90.56	89.5 (63.2–213)	0.41 ± 0.22	0.26 (0.19–0.51)	1.37 (1.1–2.22)
Female (n = 12)	245.32 ± 159.92	279.81 ± 108.92	**67.9 (49.8–89.5)**	0.34 ± 0,18	0.26 ± 0.17	1.63 (0.83–2.56)
Male (n = 12)	325.02 ± 111.59	301.46 ± 73.31	**171.4 (93.25–227.55) ***	0.49 ± 0.13	0.39 ± 0.18	1.36 (1.13–1.82)
I-III stage (n = 18)	260.58 ± 144.62	297.27 ± 96.92	88.7 (67.9–213)	0.4 ± 0.24	0.24 (0.19–0.39)	1.29 (1.07–2.18)
IV stage (n = 6)	358.93 ± 106.90	273.63 ± 74.4	98.6 (49.8–201,8)	0.44 ± 0.14	0.45 (0.2–0.59)	1.56 (1.36–3.36)
DMT2 (n = 14)	292.1 ± 148.1	283.37 ± 82.61	94.05 (70.6–213)	0.43 ± 0.24	0.37 ± 0.19	1.27 (0.67–2.26)
CVD (n = 16)	327.56 ± 129.51	300.39 ± 66.21	111.15 (69.75–218.4)	0.45 ± 0.21	0.39 ± 0.19	1.36 (0.8–2.2)

n—number; DMT2—diabetes mellitus type 2; CVD—cardiovascular diseases. Data with statistical significance are marked in bold. * *p* < 0.01: Lower serum ADAM10 concentrations in women than men; other analyzed results were not statistically significant (*p* > 0.05).

**Table 5 ijms-26-01104-t005:** NLR and PLR for the main study group. Data are expressed as mean ± SD for parametric data or as median (lower quartile–upper quartile) for nonparametric data. Means were compared using the Student’s *t*-test for normally distributed samples and the Mann–Whitney U-test for non-normally distributed samples.

Characteristic		NLR	PLR
All patients (n = 66)		3.1 (2.18–4.69)	182.89 (137–250)
Left side tumor (n = 44)		3.59 (2.29–4.7)	184.47 (143.19–250.63)
Right side tumor (n = 22)		3.3 ± 1.87	197.3 ± 108.83
I stage (n = 12)		2.45 (2.02–4.22)	156.95 (131.56–243.28)
II stage (n = 14)		3.32 (2.05–4.61)	144.52 (134.09–208.14)
III stage (n = 20)		2.69 (2.22–4.75)	237.38 ± 120.58
IV stage (n = 20)		4.13 ± 2.72	208.02 ± 103.14
DMT2 (T2DM)	Yes (n = 29)	2.7 (2.13–4.61)	157.14 (129.23–219.88)
	No (n = 37)	3.43 (2.34–4.85)	217.2 (144.33–268.06)
CVD	Yes (n = 43)	3.24 ± 1.58	**136.72 ± 67.67**
	No (n = 23)	4.03 ± 2.56	**228.77 ± 127.75 ***
Oncological treatment before surgery (RT or RCT)	Yes (n = 9)	**4.0 (3.7–4.71)**	**292.48 ± 195.5**
	No (n = 57)	**2.78 (2.14–4.68) ****	**198.8 ± 105.15 *****

n—number; DMT2—diabetes mellitus type 2; CVD—cardiovascular diseases; RT—radiotherapy; RCT—radiochemotherapy; Data with statistical significance are marked in bold. * *p* < 0.01; ** *p* < 0.05; *** *p* = 0.05. 1. PLR values were lower in patients with comorbid CVD (136.72 ± 67.67 vs. 228.77 ± 127.75; *p* = 0.007). 2. Higher values of both NLR (4.0 (3.7–4.71) vs. 2.78 (2.14–4.68); *p* = 0.041) and PLR (292.48 ± 195.5 vs. 198.8 ± 105.15; *p* = 0.05) in patients who underwent preoperative treatment. Other analyzed results were not statistically significant (*p* > 0.05).

## Data Availability

The data presented in this study are available on request from the corresponding author. The data are not publicly available due to the protection of patient data.

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
