# Peer review of "Relationship Between Systemic Inflammatory Response Exponents, Levels of ADAM10, ADAM17 Proteins and Selected Clinical Parameters in Patients with Colorectal Cancer: Original Research Study"

_ijms, 2025, doi:10.3390/ijms26031104_

Round 1
Reviewer 1 Report
Comments and Suggestions for Authors
The manuscript written by Magdalena Sikora-Skrabaka and colleagues examined the relationship between systemic inflammatory response (SIR) indicators, such as neutrophil-to-lymphocyte ratio (NLR) and platelet-to-lymphocyte ratio (PLR), and the molecular markers ADAM10 and ADAM17 in colorectal cancer (CRC) patients. Tumor tissue, surgical margins, and blood from 66 CRC patients were analyzed. Results showed that PLR was lower in patients with cardiovascular disease (CVD) and higher in those who underwent preoperative treatment. No clear correlations were found between ADAM10 and ADAM17 levels and SIR parameters. The study suggests that comorbidities and preoperative treatment should be considered when assessing SIR in CRC patients.
The manuscript is well written, but I believe some minor revisions are needed:
- The authors should analyze the levels of ADAM10 and ADAM17 in the histological samples from CRC patients using more specific techniques, such as Western blotting, immunohistochemistry, or immunofluorescence.
- It has been reported that L1CAM expression correlates with ADAM10 levels in several tumors, including CRC (doi:10.1158/0008-5472.CAN-07-0991, doi:10.2147/IJN.S480168), and is associated with a worse prognosis. The authors have to analyze L1CAM concentrations in both tissue and blood serum samples from CRC patients and cite these articles.
good
Reviewer 2 Report
Comments and Suggestions for Authors
1. Abbreviations and Acronyms:
Abbreviations should be defined upon their first appearance in each section: Abstract, Main Text, and the first
figure or table.
DMT2: Define in (Lines 33, 75, Table 1).
CVD: Define in (Line 76, Table 1).
n: Define in (Table 1).
RT and RCT: Define in (Table 5).
Change TNFalpha to TNF-alpha and define (Lines 211, 275).
ELISA: Define in (Line 27).
Correct "to" to "two" or remove it (Line 260).
Define IGFs, VEGF, FGF, EGF (Line 275).
Correct grammar: "NLR and PLR were assessed" (Line 320).
Write cm³ correctly (Lines 313, 365).
Replace "or" with "and" after "normal distribution" (Line 389).Correct "i" to "in" (Figure 1, Line 160).
Add "were collected" after 1 cm³ (Line 313).
Use ELISA instead of its full form (Line 347).
2. Title:
Include the study type in the title for clarity and precision.
3. Materials and Methods:
Add clear section headings:
4.1 Study Population
4.2 Total Protein Concentration in Tissue Samples, and so forth.
Properly cite manufacturers and suppliers, e.g., (PRO Scientific Inc, Oxford, CT, USA), (PBS without Ca and
Mg; pH 7.4; BIOMED, Lublin, Poland), and (Sigma-Aldrich, St. Louis, MO, USA).
Specify the software version used (e.g., Data Analysis Software Kc Junior) (Lines 351, 359).
Correct "Serum samples were frozen at -80°C" (Line 366).
Specify the antibodies used for ADAM10 and ADAM17 (Lines 368-369).
Clarify the purpose of using Biotin-avidin and horseradish peroxidase (Line 369).
Correct to Shapiro-Wilk W test (Line 387).
Specify variables deviating from normal distribution (Lines 389, 392).
Remove sections 4.6 (Research Hypothesis) and 4.7 (Aim of the Study) from Methods.
4. Results:
Improve Tables 2, 4, and 5:
Add p-values to Table 2.
If results are not statistically significant, state this in the footnote (p > 0.05).
Classify DMT2 and CVD as Yes/No in Table 2 and 4, similar to Table 5.
Add a note to Tables 4 and 5 clarifying that other analyzed results were not statistically significant (p > 0.05).
Remove or rephrase redundant text (Lines 139-141, 175-177, 182-183).
Improve Figure 3 to ensure it matches the clarity and structure of Figures 1 and 2.5. Discussion:
Add a clear limitations section to provide context to the study's constraints and scope.
6. Conclusions:
Rewrite the Conclusions for clarity and better alignment with the study results.
Clarify whether the systemic inflammatory response (SIR) increases CRC risk, as implied in various sections.
Resolve the inconsistency regarding ADAM10 and ADAM17's role in inflammatory pathways across the
Conclusions, Discussion (Lines 195-197, 283-284), and Results (Lines 154-157).
Remove Lines 414-418, as limitations should be discussed in the Discussion section.
7. Additional Notes:
Improve Figures 1-3 for clarity and visual appeal.
Include Figures for all results, even if not statistically significant (e.g., ADAM10 concentration in serum and
tumor samples, Lines 184-192).
If including all figures significantly lengthens the manuscript, mo
Comments on the Quality of English LanguageEnglish can be improved but not too poor.
Reviewer 3 Report
Comments and Suggestions for Authors
Dear authors,
Your manuscript 'Relationship between systemic inflammatory response exponents, levels of ADAM10, ADAM17 proteins and selected clinical parameters in patients with colorectal cancer' presents results of measuring ADAM10/17 levels in tumor and normal-adjacent tissues and peripheral blood of patients with colorectal cancer (CRC). In addition, neutrophil-to-lymphocyte (NLR) and platelet-to-lymphocyte (PLR) ratios were estimated. After comparing these biomarkers with the clinical data of these patients, you proposed that PLR and/or NLR could be used to differentiate patients with comorbidities or who underwent different treatments. Though the potential contribution of this study to their topic, I would like to comment on some concerns.
Major comments
1. Please add a statement regarding the statistical power of your sample. Justify the reasons for setting a significance level of 0.05 (instead of 0.01) for further analyses.
2. In Figure 3, there is a missing panel. Please verify it.
3. This is required to use specific methods to validate your findings. For example, you should use Pearson's correlation test to compare quantitative variables (instead of Spearman's test).
4. There is no comparison between normal-tumor ADAM10/17 levels. How informative are these markers? Do they be considered liquid biopsy markers by following their blood levels? Please evaluate these topics and discuss the results accordingly.
Minor comments
5. Please describe in detail all figures and tables in their legends and footnotes. All visual elements should be self-explanatory.
Round 2
Reviewer 2 Report
Comments and Suggestions for Authors
Thank you for taking all of my comments into consideration!
Reviewer 3 Report
Comments and Suggestions for Authors
Dear authors,
Your manuscript 'Relationship between systemic inflammatory response exponents, levels of ADAM10, ADAM17 proteins and selected clinical parameters in patients with colorectal cancer' presents results of measuring ADAM10/17 levels in tumor and normal-adjacent tissues and peripheral blood of patients with colorectal cancer (CRC). In addition, neutrophil-to-lymphocyte (NLR) and platelet-to-lymphocyte (PLR) ratios were estimated. After comparing these biomarkers with the clinical data of these patients, you proposed that PLR and/or NLR could be used to differentiate patients with comorbidities or who underwent different treatments. Though the potential contribution of this study to their topic, I felt authors' responses did not address my concerns properly. For example,
1. I requested information about the statistical power of the sample. It is not clear how representative are the 24 patients included in this study in comparison with all potential patients with colorectal cancer in your hospital.
2. In your response, you described many outliers in your data. Did you find any potential cause to them? It should be further discussed. Following my previous comment, it does not seem to be a clear representation of the colorectal cancer population.
3. As you stated, "far-reaching conclusions cannot be drawn", therefore I suggest to strength the validation, explanation, and discussion of your findings. In their current form, it could be speculative. How can we expect to have different ADAM 10/17 leves in blood without tumor alterations of these markers in the tumors?
Round 3
Reviewer 3 Report
Comments and Suggestions for Authors
Dear authors,
Your manuscript 'Relationship between systemic inflammatory response exponents, levels of ADAM10, ADAM17 proteins and selected clinical parameters in patients with colorectal cancer' presents results of measuring ADAM10/17 levels in tumor and normal-adjacent tissues and peripheral blood of patients with colorectal cancer (CRC). In addition, neutrophil-to-lymphocyte (NLR) and platelet-to-lymphocyte (PLR) ratios were estimated. After comparing these biomarkers with the clinical data of these patients, you proposed that PLR and/or NLR could be used to differentiate patients with comorbidities or who underwent different treatments. Thank you for having addressed my previous comments.